# Comparative Analysis of da Vinci^®^ Xi and hinotori™ SRS Robot-Assisted Surgery Systems for Gynecologic Disorders: A Retrospective Study

**DOI:** 10.3390/medicina60122014

**Published:** 2024-12-06

**Authors:** Shinichi Togami, Nozomi Furuzono, Yusuke Kobayashi, Chikako Nagata, Mika Fukuda, Mika Mizuno, Shintaro Yanazume, Hiroaki Kobayashi

**Affiliations:** Department of Obstetrics and Gynecology, Faculty of Medicine, Kagoshima University, Kagoshima 890-8520, Japan; nozomi.fur051120@gmail.com (N.F.); ykobayashi@kufm.kagoshima-u.ac.jp (Y.K.); nagata1015.kag@gmail.com (C.N.); sakura.mika1014@icloud.com (M.F.); mizunomizuno@kufm.kagoshima-u.ac.jp (M.M.); s-yana@m3.kufm.kagoshima-u.ac.jp (S.Y.); hirokoba@m2.kufm.kagoshima-u.ac.jp (H.K.)

**Keywords:** da Vinci^®^ Xi, gynecologic diseases, hinotori^TM^, robotic surgery, surgical robot system

## Abstract

*Background and Objectives*: This study aims to evaluate and compare the safety and efficacy of the da Vinci^®^ Xi and hinotori™ SRS robot-assisted surgical systems for gynecologic disorders. *Materials and Methods*: We conducted a retrospective study of 401 cases (43 benign uterine tumors; 88 pelvic organ prolapses; 270 low-risk endometrial cancers) of robot-assisted surgery performed at Kagoshima University Hospital between January 2017 and October 2024. Surgical factors such as the operative time, blood loss, and complication rates were analyzed and compared between the da Vinci^®^ Xi (332 cases) and hinotori™ SRS (69 cases) systems. Complications were classified according to the Clavien–Dindo classification, with Grade 2 or higher considered significant. *Results*: Significant differences were observed between the two groups in terms of age, body mass index, cockpit/console time, and median time from roll-in to cockpit/console start. The cockpit/console time was significantly longer for the hinotori™ SRS system (173 min) compared to the da Vinci^®^ Xi (156 min; *p* = 0.047). No significant differences were observed in the total operative time, blood loss, or length of hospital stay. Intraoperative complications were minimal, with one case of bladder injury and one case of vascular injury recorded for the da Vinci^®^ Xi. The overall postoperative complication rate was approximately 4%. *Conclusions*: Robot-assisted surgery using both the da Vinci^®^ Xi and hinotori™ SRS systems was found to be safe, with minimal blood loss and a low complication rate. The hinotori™ SRS system demonstrated operative outcomes comparable to those of the da Vinci Xi^®^ system, suggesting that it may serve as a viable alternative. Further prospective studies are warranted to evaluate the efficacy and safety of these systems.

## 1. Introduction

Robot-assisted surgery was first introduced into surgical medicine in the 1990s and has proliferated rapidly alongside technological advancements, expanding its application across multiple fields. The da Vinci^®^ system, developed in the United States and made commercially available in the early 2000s, has played a groundbreaking role as an alternative to laparoscopic surgery in various surgical fields. The da Vinci^®^ system offers advantages such as multi-jointed arms that enable precise manipulation, three-dimensional high-resolution visualization, and computer-controlled accuracy in replicating the surgeon’s movements, which have established it as a standard approach in the treatment of benign and malignant tumors in gynecology.

Robot-assisted surgery for gynecologic procedures places less burden on patients than traditional open and laparoscopic surgery [1,2,3]. For instance, open surgery is typically more invasive and involves longer postoperative hospital stays, whereas robotic surgery requires only small incisions, allowing for reduced postoperative pain, shorter hospital stays, and minimal scarring. Additionally, three-dimensional visualization and precise multi-joint arms allow for the reduction in blood loss and shorter operative times in gynecologic procedures that require fine manipulation. Furthermore, robot-assisted surgery minimizes the effects of hand tremors, reduces surgeon fatigue, and facilitates more stable procedures.

However, robotic surgery has several disadvantages. Among these is the high cost of installation, maintenance, and training for robotic surgery systems [4]. Additionally, robot-assisted surgery is generally more complex than traditional laparoscopic surgery, resulting in a steep learning curve for surgeons and the risk of extended surgery time for those with limited experience. Furthermore, the need for a backup system to prepare for potential mechanical failures places a considerable burden on institutions that implement these systems.

In Japan, the presence of a national health insurance system means that surgical procedures not covered by insurance tend to see limited adoption. However, the inclusion of robotic surgery for gynecologic procedures under insurance coverage in 2018 led to a rapid increase in the number of robotic surgeries performed nationwide [5]. In recent years, new robot-assisted surgical systems have been developed, expanding the range of available options beyond the conventional da Vinci system [6,7]. In Japan, a domestically produced robotic surgery system, hinotori, was developed and approved for gynecological use in 2022. The hinotori™ SRS surgical robot system offers a more cost-effective option than the da Vinci system, making it a promising choice for adoption in Japanese hospitals. We have previously reported the safety of the hinotori™ SRS based on 12 cases of gynecological disorders [8]. The hinotori™ SRS is an integrated surgical robotic system similar to the da Vinci^®^ Xi, featuring robotic arms with eight axes designed to closely mimic human movements. A distinguishing characteristic of the system is that the robotic arms do not dock with the trocars and instead use a pivot as the fulcrum for movement [8]. As of the end of January 2024, hinotori™ SRS has been approved in two countries, and 45 hinotori™ SRS have been introduced in Japan [6].

The advent and evolution of robotic surgery have not only broadened the treatment options for gynecologic disorders but also raised significant expectations regarding its safety and therapeutic effectiveness. However, as this technology remains relatively new, evidence concerning its safety and efficacy is yet to be fully established. Specifically, it is essential to elucidate the impact of these new surgical robots on patient safety and compare them with the da Vinci system in terms of their advantages and potential challenges.

This study aimed to evaluate robot-assisted surgery for gynecological disorders based on surgical data from a single institution, with a particular focus on assessing the safety and efficacy of the new hinotori™ SRS system.

## 2. Materials and Methods

### 2.1. Patients

This retrospective study included 401 patients who underwent robot-assisted surgery at the Kagoshima University Hospital between January 2017 and October 2024. The surgeries were conducted using either the da Vinci^®^ Xi (Intuitive Surgical, Sunnyvale, CA, USA) or hinotori™ SRS systems (Medicaroid Corporation, Kobe, Japan).

### 2.2. Robotic Surgery Procedure

The choice of robotic system was determined by the time of system implementation and the availability of each platform (Table 1). We began performing surgeries using the da Vinci^®^ Xi system in 2017, prior to the approval of robotic surgery for insurance coverage in Japan. In 2018, low-risk endometrial cancer (defined as less than 50% myometrial invasion and endometrioid carcinoma Grade 1 or 2) and benign uterine tumors (such as uterine myoma, adenomyosis, and high-grade squamous intraepithelial lesion) became eligible for insurance coverage. Furthermore, pelvic organ prolapse (POP) was added to the list of covered indications in 2020. The hinotori™ SRS, a Japanese-made robotic surgical system, received insurance approval in December 2022, allowing its use for low-risk endometrial cancer, benign uterine tumors, and POP.

Although there is a significant difference in the total number of surgeries performed with each robotic system due to their respective timelines, there is no difference in the surgical indications between the two systems. Additionally, at our institution, pre-determined surgical slots are allocated specifically for the da Vinci^®^ Xi system and hinotori™ SRS, and cases of low-risk endometrial cancer, benign uterine tumors, and POP were sequentially assigned to these slots. Therefore, there was no patient allocation based on the type of robotic system used.

The surgical protocols for da Vinci^®^ Xi and hinotori™ SRS, including anesthesia, patient positioning, and trocar placement, followed the same standardized protocol. General anesthesia was administered without epidural anesthesia. Patients were positioned in the lithotomy position, with four robotic trocars placed 7–9 cm apart at the level of the umbilicus, and the assistant port was positioned at the left end. Once port placement was completed, the patient was placed in a 27° Trendelenburg position, and roll-in was performed. After the insertion of all robotic instruments, the console/cockpit time began. Simple hysterectomy and bilateral salpingo-oophorectomy were performed for benign uterine tumors. Sacrocolpopexy (RSC) was performed for POP, while simple hysterectomy, bilateral salpingo-oophorectomy, and sentinel node navigation surgery were performed for low-risk endometrial cancer. The surgeries were performed by four gynecologic oncologists.

### 2.3. Evaluation of Surgical Factors

The surgical factors evaluated in this study included patient age, body mass index (BMI), total operative time, console/cockpit time, time from surgery initiation to roll-in, time from roll-in to the start of robotic surgery, estimated blood loss, days to postoperative discharge, conversion rate to open surgery, and rates of intraoperative and postoperative complications. Intraoperative complications included bowel, bladder, ureter, or blood vessel injuries. The Clavien–Dindo classification is a widely used system for categorizing surgical complications based on their severity. It ranges from Grade I, which includes minor complications requiring no treatment or minimal intervention, to Grade V, which indicates patient death. Grades II through IV involve increasing levels of medical intervention, ranging from pharmacological treatments to life-threatening complications requiring intensive care. Postoperative complications were classified according to the Clavien–Dindo classification system, with Grade 2 or higher considered significant.

### 2.4. Statistical Analysis

Comparative statistical analysis of surgical factors related to the da Vinci^®^ Xi and hinotori™ SRS systems was conducted using the chi-square test and the Wilcoxon signed-rank test. The JMP software (version 14, SAS Institute Inc., Tokyo, Japan) was used for the analysis.

## 3. Results

Figure 1 shows the annual trend in the number of robotic surgeries performed for benign uterine tumors, POP, and low-risk endometrial cancer in the Department of Obstetrics and Gynecology at Kagoshima University Hospital. To date, 332 surgeries have been performed using the da Vinci^®^ Xi system and 69 using the hinotori™ SRS, with a consistent annual increase in the total number of robotic surgeries. Although the hinotori™ SRS was only introduced in 2022, the number of cases has steadily increased each year. Figure 2 shows the annual distribution of robotic surgeries for target diseases. As of October 2024, only gynecological procedures for benign uterine tumors, POP, and low-risk endometrial cancer are covered by insurance in Japan, with low-risk endometrial cancer surgeries and RSC accounting for approximately 85% of all cases.

Figure 3 depicts the number of surgeries for each condition performed using the da Vinci^®^ Xi and hinotori™ SRS systems. Robotic surgeries were performed in 43 cases of benign uterine tumors, 88 cases of POP, and 270 cases of low-risk endometrial cancer using the hinotori™ SRS system used for surgeries across all conditions. Table 2 presents a comparison of surgical factors between the da Vinci^®^ Xi and hinotori™ SRS systems. Significant differences were observed between the two groups in terms of age, BMI, cockpit/console time, and median time from roll-in to cockpit/console initiation. However, no significant differences were found in the total operative time, median time from the start of the operation to roll-in, estimated blood loss, or postoperative length of stay. There were also no significant differences in the intraoperative and postoperative complication rates between the two groups. Intraoperative complications were observed in two cases (1%) with the da Vinci^®^ Xi system, including one case of external iliac vein injury and one case of bladder injury. Postoperative complications were observed in 13 cases with the da Vinci^®^ Xi system, including pelvic infection (8 cases), ileus (2 cases), vaginal cuff hematoma (2 cases), and ureteral stricture (1 case). In contrast, postoperative complications with the hinotori™ SRS included pelvic infection in two cases.

## 4. Discussion

In Japan, robot-assisted surgery for low-risk endometrial cancer and benign uterine tumors was approved for insurance coverage in 2018, and in 2020, RSC for POP was also included. At our institution, robot-assisted surgery using the da Vinci^®^ Xi system was initiated in 2017, with the hinotori™ SRS system implemented in 2022. To date, 332 cases have been performed using the da Vinci^®^ Xi and 69 cases with the hinotori™ SRS, with a steady annual increase in the total number of robotic surgeries in our institute. Robotic surgery is now widely adopted and continues to rise globally [9], with reports indicating over 50,000 certified surgeons and more than 6730 robotic systems across 69 countries [10]. In Japan, a survey conducted by Komatsu et al., comprising members of The Japan Society of Gynecologic and Obstetric Endoscopy, reported that 38% (333/870) of respondents had experience in robotic surgery [11]. Since our initial report on the use of the hinotori™ SRS for gynecologic disorders, numerous reports have emerged detailing the use of the hinotori™ SRS across various fields, with the number of such cases increasing each year [6,12,13,14,15]. In our initial report, we presented 12 cases of hinotori SRS used for gynecological disorders; however, since 2022, the number of cases has increased annually, with a total of 69 cases performed to date. This number is expected to continue increasing.

Among various diseases, low-risk endometrial cancer has the highest number of cases, with an increasing trend observed each year. For low-risk endometrial cancer, we perform sentinel node (SN) navigation surgery, omitting systematic pelvic lymphadenectomy in cases where SN metastasis is negative [16]. To identify the SNs, we use a hybrid method combining radioisotope and indocyanine green tracers, both of which are compatible with the da Vinci^®^ Xi and hinotori™ SRS systems. In a randomized controlled trial comparing robot-assisted surgery with conventional laparoscopic surgery for endometrial cancer, a slight increase in overall survival was observed using robot-assisted surgery compared to conventional laparoscopic surgery. Therefore, robot-assisted surgery for endometrial cancer has been suggested as safe [17].

With recent advancements in robotic surgery, the use of robotic RSC for the surgical treatment of POP has increased. A systematic review comparing RSC with laparoscopic sacrocolpopexy (LSC) reported that RSC tends to result in less blood loss but longer operative times than in LSC, whereas the recurrence rates of POP were found to be equivalent between the two approaches [2]. Ichino et al. published the first report on RSC using the hinotori™ SRS, indicating that the hinotori™ SRS demonstrated perioperative outcomes nearly equivalent to those of the da Vinci^®^ system [12].

This study represents the largest analysis to date comparing surgical factors between the da Vinci^®^ Xi and hinotori™ SRS systems for gynecologic disorders. The cockpit/console times were 156 min for da Vinci^®^ Xi and 173 min for hinotori™ SRS, indicating a significantly longer time for hinotori™ SRS (*p* = 0.047). One possible reason for this is that approximately 35% of hinotori SRS cases involved RSC, a procedure with longer operative times than other techniques. Additionally, the median time from roll-in to cockpit/console start was significantly longer for the hinotori™ SRS, likely because of the smaller number of cases, which may reflect the impact of the learning curve for this system.

Motoyama et al. [18] analyzed perioperative outcomes of partial nephrectomy performed using the da Vinci^®^ Xi and hinotori™ SRS systems, reporting no significant differences in the operative time or complications. These findings are consistent with our results. In this study, no significant differences were observed in intraoperative or postoperative complications between the da Vinci^®^ Xi and hinotori™ SRS systems. Petersen et al. [19] reported that the risk of urological injury in robot-assisted hysterectomies was 0.92%. Similarly, in our study, the only intraoperative complications observed with the da Vinci Xi system were one case of bladder injury and one case of vascular injury. Balafoutas et al. [20] reported a postoperative complication rate of 10.9% for Clavien–Dindo Grade 2 or higher in 110 cases of robotic surgery for gynecologic disorders. In contrast, our study showed a lower postoperative complication rate of approximately 4%.

The limitations of this study include the small sample size of the hinotori SRS group, which may have made precise comparisons between the two groups challenging. Secondly, this was a retrospective study.

## 5. Conclusions

The use of the da Vinci^®^ Xi system and hinotori™ SRS at our institution has shown a yearly increasing trend, with a particularly rapid rise in the number of cases following their inclusion under insurance coverage. Although the hinotori™ SRS has only recently been introduced in Japan and the number of cases remains limited, robot-assisted surgery at our institution was performed safely with minimal blood loss and few major intraoperative and postoperative complications. Furthermore, the findings suggest that the hinotori™ SRS system can achieve comparable operative times and blood loss levels to those of the da Vinci^®^ Xi for gynecologic procedures. Therefore, we believe that gynecological surgeries performed using the hinotori™ SRS system can provide perioperative outcomes that are non-inferior to those achieved with the da Vinci^®^ Xi system. Future prospective studies with larger sample sizes are needed to evaluate the efficacy and safety of the da Vinci^®^ Xi and hinotori™ SRS systems.

## Figures and Tables

**Figure 1 medicina-60-02014-f001:**
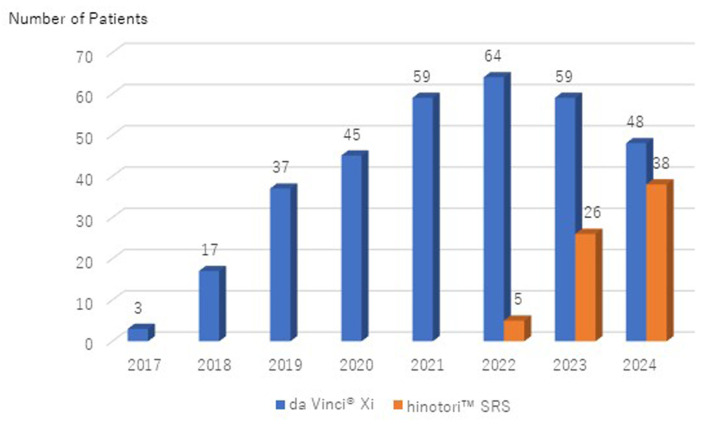
Annual trend in number of robotic surgeries at Kagoshima University Hospital.

**Figure 2 medicina-60-02014-f002:**
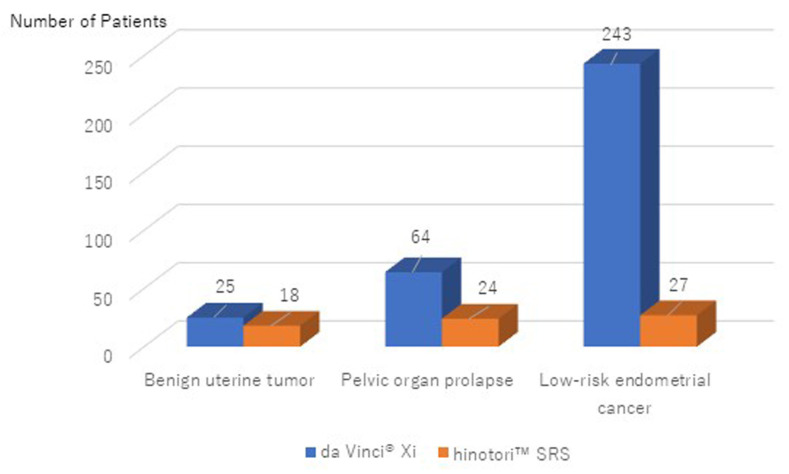
Number of robotic surgeries by disease at Kagoshima University Hospital.

**Figure 3 medicina-60-02014-f003:**
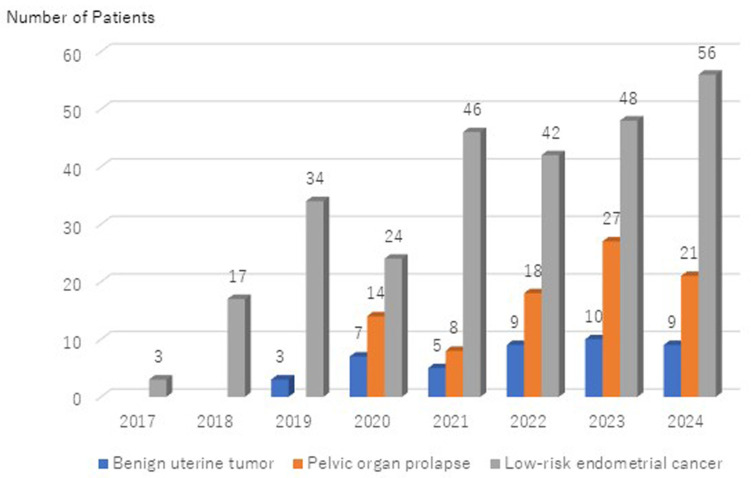
Annual trend in gynecologic diseases targeted for robotic surgery at Kagoshima University Hospital.

**Table 1 medicina-60-02014-t001:** Annual trends in surgical indications for the da Vinci^®^ Xi system and hinotori™ SRS.

Year	da Vinci^®^ Xi	hinotori™ SRS
2017	(da Vinci^®^ Xi introduced) Low-risk endometrial cancer (one’s own expense)	
2018	Low-risk endometrial cancer (health insurance)	
2019	Benign uterine tumor Low-risk endometrial cancer (health insurance)	
2020	Benign uterine tumor Low-risk endometrial cancer Pelvic organ prolapse (health insurance)	
2021	Benign uterine tumor Low-risk endometrial cancer Pelvic organ prolapse (health insurance)	
2022	Benign uterine tumor Low-risk endometrial cancer Pelvic organ prolapse (health insurance)	(hinotori™ SRS introduced) Benign uterine tumor Low-risk endometrial cancer Pelvic organ prolapse (health insurance)
2023	Benign uterine tumor Low-risk endometrial cancer Pelvic organ prolapse (health insurance)	Benign uterine tumor Low-risk endometrial cancer Pelvic organ prolapse (health insurance)
2024	Benign uterine tumor Low-risk endometrial cancer Pelvic organ prolapse (health insurance)	Benign uterine tumor Low-risk endometrial cancer Pelvic organ prolapse (health insurance)

**Table 2 medicina-60-02014-t002:** Comparison of the clinicopathological characteristics between hinotori SRS and da Vinci Xi.

Characteristic	da Vinci^®^ Xi (*n* = 332)	hinotori™ SRS (*n* = 69)	*p*-Value
Median age (years) Median BMI (kg/m^2^) Median operation time (min)	58 (28–89)	64 (28–86)	0.019
	27 (17.3–53.1)	25.4 (17.3–43.3)	0.049
	213 (68–638) (95% CI: 210.4–227.6)	238 (92–612) (95% CI: 215.0–255.1)	0.075
Median cockpit/console time (min) Median time from operation start to roll-in (min)	156 (53–520) (95% CI: 158.1–172.8)	173 (60–430) (95% CI: 164.0–195.9)	0.047
	17 (6–119) (95% CI: 18.9–21.5)	16 (7–114) (95% CI: 15.5–21.8)	0.2
Median time from roll-in to cockpit/console start (min) Median blood loss (mL) Median length of hospital stay (days) Diagnosis	9 (5–50) (95% CI: 9.9–11.0)	13 (5–36) (95% CI: 12.4–15.1)	<0.0001
	21 (1–1125)	23 (3–226)	0.98
	6 (3–68)	6 (3–22)	0.15
			<0.0001
Benign uterine tumor Pelvic organ prolapse	25 (8%)	18 (26%)	
	64 (19%)	24 (35%)	
Low-risk endometrial cancer	243 (73%)	27 (39%)	
Conversion to open surgery			
Yes	0 (0%)	0 (0%)	
No	332 (100%)	69 (100%)	
Intraoperative complications			0.51
Yes	2 (1%)	0 (0%)	
No	330 (99%)	69 (100%)	
Postoperative complications			0.69
Yes	13 (4%)	2 (3%)	
No	319 (96%)	67 (97%)	

Data are presented as the median (range) or *n* (%). BMI: body mass index. Statistical significance of differences between groups was tested using the x^2^ test for categorical data and the Wilcoxon signed-rank test for continuous parameters.

## Data Availability

The data presented in this study are openly available in Kagoshima university at https://www.obgy-kagoshima.jp/ (4 December 2024).

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
