# Peer review of "Comparative Analysis of da Vinci^®^ Xi and hinotori™ SRS Robot-Assisted Surgery Systems for Gynecologic Disorders: A Retrospective Study"

_medicina, 2024, doi:10.3390/medicina60122014_

Round 1
Reviewer 1 Report
Comments and Suggestions for Authors
The authors tried to compare the daVinci Xi and Hinotori robotic systems. However, the year in which the two robotic systems were introduced and the indications for surgery by each robotic system are quite different. The authors should make another table showing all the variables in each procedure.
There are several questions about this manuscript.
1. How did you select the patients for each robotic system?
2. What was the indication of the benign uterine tumor?
3. What is the definition of low-risk endometrial cancer? Why was Hinotori SRS not chosen for surgery on low-risk endometrial cancer?
4. Specify the intra- and postoperative complications in detail.
Author Response
|
Comments 1: The authors tried to compare the daVinci Xi and Hinotori robotic systems. However, the year in which the two robotic systems were introduced and the indications for surgery by each robotic system are quite different. The authors should make another table showing all the variables in each procedure.
|
|
Response 1: Thank you for pointing this out. We agree with this comment. Therefore, we provide a clearer representation of the year each robotic system was introduced and their respective surgical indications, the annual trends in surgical indications for procedures performed using the da Vinci® Xi system and hinotori™ SRS have been added as Table 1.
|
|
Comments 2: How did you select the patients for each robotic system? |
|
Response 2: Thank you for your valuable comment. We began performing surgeries using the da Vinci® Xi system in 2017, prior to the approval of robotic surgery for insurance coverage in Japan. In 2018, low-risk endometrial cancer (defined as less than 50% myometrial invasion and endometrioid carcinoma grade 1 or 2) and benign uterine tumors (such as uterine myoma, adenomyosis, and HSIL) became eligible for insurance coverage. Furthermore, pelvic organ prolapse (POP) was added to the list of covered indications in 2020. The hinotori™ SRS, a Japanese-made robotic surgical system, received insurance approval in December 2022, allowing its use for low-risk endometrial cancer, benign uterine tumors, and POP. Although there is a significant difference in the total number of surgeries performed with each robotic system due to their respective timelines, there is no difference in the surgical indications between the two systems. Additionally, at our institution, pre-determined surgical slots are allocated specifically for the da Vinci® Xi system and hinotori™ SRS, and cases of low-risk endometrial cancer, benign uterine tumors, and POP were sequentially assigned to these slots. Therefore, there was no patient allocation based on the type of robotic system used. Based on the reviewers' feedback, we have added this information to the material and methods section. (Line 91-107)
|
|
Comments 3: What was the indication of the benign uterine tumor? |
|
Response 3: Thank you for your valuable comment. The indications for benign uterine tumors include uterine myoma, uterine adenomyosis, and high-grade squamous intraepithelial lesion (HSIL). Based on the reviewers' feedback, we have added this information to the material and methods section. (Line 95-97). |
|
Comments 4: What is the definition of low-risk endometrial cancer? Why was Hinotori SRS not chosen for surgery on low-risk endometrial cancer? |
|
Response 4: Thank you for your valuable comment. Low-risk endometrial cancer is defined as less than 50% myometrial invasion with endometrioid carcinoma grade 1 or 2. As shown in Table 2, 27 cases of surgery for low-risk endometrial cancer were performed using the hinotori™ SRS. Patient allocation was not influenced by the type of robotic system used. Based on the reviewers' feedback, we have added this information to the material and methods section. (Line 94-95). |
|
Comments 5: Specify the intra- and postoperative complications in detail. |
|
Response 5: Thank you for your valuable comment. Intraoperative complications were observed in 2 cases (1%) with the da Vinci® Xi system, including one case of external iliac vein injury and one case of bladder injury. Postoperative complications were observed in 13 cases with the da Vinci® Xi system, including pelvic infection (8 cases), ileus (2 cases), vaginal cuff hematoma (2 cases), and ureteral stricture (1 case). In contrast, postoperative complications with the hinotori™ SRS included pelvic infection in 2 cases. Based on the reviewers' feedback, we have added this information to the results section. (Line165-170). |

Reviewer 2 Report
Comments and Suggestions for Authors
The study titled “Comparative Analysis of da Vinci® Xi and hinotori™ SRS Robot-Assisted Surgery Systems for Gynecologic Disorders: A Retrospective Study” presents a relevant analysis of two robotic surgery systems, da Vinci® Xi and hinotori™ SRS, in the context of gynecologic disorders. The topic is timely and significant, given the increasing adoption of robotic technologies in minimally invasive surgeries. The study effectively highlights the efficacy, safety, and limitations of both systems, making a valuable contribution to the literature, particularly considering the recent introduction of hinotori™ in Japan.
The manuscript has several strengths, including the relevance of the topic, a well-described methodology, and a significant scientific contribution by addressing a gap in the literature. It is the largest comparative study of these two systems in gynecology to date. Moreover, the results are clearly presented, with tables and figures that facilitate understanding. However, some improvements could be made to enhance the article’s scientific impact.
In the introduction, the contextualization is adequate, but additional information about the specific technical differences between the systems would be beneficial. This could include the technical design of hinotori™ SRS, its cost-effectiveness, and its adaptations to the Japanese market, helping to frame the factors influencing the outcomes. In the methodology, while the inclusion criteria and standardized surgical procedures are well-described, the small number of cases in the hinotori™ SRS group represents a significant limitation and should be emphasized in the discussion. Additionally, a brief explanation of the use of the Clavien-Dindo classification for surgical complications would help readers less familiar with the methodology.
The results are clear but could benefit from additional analyses, such as confidence intervals for operative times and complications, to strengthen statistical rigor. In the discussion, it would be helpful to expand comparisons with the literature, exploring how the findings relate to other studies. The authors could also highlight practical implications, such as the cost-effectiveness and learning curve of the hinotori™ SRS. Including a more detailed analysis of the sample size limitations and their impact on the results would further strengthen this section.
The conclusion is consistent with the data presented but could include clearer recommendations regarding the practical feasibility of hinotori™ SRS compared to da Vinci® Xi. Emphasizing the need for prospective studies with larger sample sizes is also essential to validate the findings.
From a technical perspective, the tables and figures are well-organized, but some legends could be more detailed. References should be reviewed to ensure completeness and adherence to the required format. Overall, the text is well-written, but improving the flow of some paragraphs, especially in the discussion, would make the manuscript more cohesive and objective.
In conclusion, the study is a valuable contribution to the field, addressing a highly relevant topic. With the suggested revisions, the manuscript could have an even greater scientific and practical impact. Congratulations to the authors for the execution of this study and their choice of topic.
Author Response
|
Comments 1: In the introduction, the contextualization is adequate, but additional information about the specific technical differences between the systems would be beneficial. This could include the technical design of hinotori™ SRS, its cost-effectiveness, and its adaptations to the Japanese market, helping to frame the factors influencing the outcomes.
|
|
Response 1: Thank you for pointing this out. We agree with this comment. The hinotori™ SRS is an integrated surgical robotic system similar to the da Vinci® Xi, featuring robotic arms with 8 axes designed to closely mimic human movements. A distinguishing characteristic of the system is that the robotic arms do not dock with the trocars and instead use a pivot as the fulcrum for movement. As of the end of January 2024, hinotori™ SRS has been approved in two countries, and 45 hinotori™ SRS have been introduced in Japan. Based on the reviewers' feedback, we have added this information to the material and methods section. (Line 69-74) |
|
Comments 2: In the methodology, while the inclusion criteria and standardized surgical procedures are well-described, the small number of cases in the hinotori™ SRS group represents a significant limitation and should be emphasized in the discussion. Additionally, a brief explanation of the use of the Clavien-Dindo classification for surgical complications would help readers less familiar with the methodology. |
|
Response 2: Thank you for your valuable comment. We began performing surgeries using the da Vinci® Xi system in 2017, prior to the approval of robotic surgery for insurance coverage in Japan. In 2018, low-risk endometrial cancer (defined as less than 50% myometrial invasion and endometrioid carcinoma grade 1 or 2) and benign uterine tumors (such as uterine myoma, adenomyosis, and HSIL) became eligible for insurance coverage. Furthermore, pelvic organ prolapse (POP) was added to the list of covered indications in 2020. The hinotori™ SRS, a Japanese-made robotic surgical system, received insurance approval in December 2022, allowing its use for low-risk endometrial cancer, benign uterine tumors, and POP. Thus, additional information was included to highlight the limited number of cases due to the recent introduction of the hinotori™ SRS. The Clavien-Dindo classification is a widely used system for categorizing surgical complications based on their severity. It ranges from Grade I, which includes minor complications requiring no treatment or minimal intervention, to Grade V, which indicates patient death. Grades II through IV involve increasing levels of medical intervention, ranging from pharmacological treatments to life-threatening complications requiring intensive care. Based on the reviewers' feedback, we have added this information to the material and methods section. (Line 91-100 and 125-130)
|
|
Comments 3: The results are clear but could benefit from additional analyses, such as confidence intervals for operative times and complications, to strengthen statistical rigor. In the discussion, it would be helpful to expand comparisons with the literature, exploring how the findings relate to other studies. The authors could also highlight practical implications, such as the cost-effectiveness and learning curve of the hinotori™ SRS. Including a more detailed analysis of the sample size limitations and their impact on the results would further strengthen this section. |
|
Response 3: Thank you for your valuable comment. In response to the reviewer’s comments, we have added 95% confidence intervals (CIs) for operative time and cockpit/console time. Additionally, in the Discussion section, we have incorporated further analysis referencing a study that compared perioperative outcomes of partial nephrectomy using the da Vinci® Xi and hinotori™ SRS systems. Regarding the learning curve for robot-assisted radical prostatectomy, it has been reported that 50–100 cases are required for a single surgeon to achieve stable operative times. However, in this study, the number of cases performed by a single surgeon did not exceed 30, making it difficult to analyze the learning curve. Based on the reviewers' feedback, we have added this information to the discussion section. (table 2 and Line 222-224). |
|
Comments 4: The conclusion is consistent with the data presented but could include clearer recommendations regarding the practical feasibility of hinotori™ SRS compared to da Vinci® Xi. Emphasizing the need for prospective studies with larger sample sizes is also essential to validate the findings. |
|
Response 4: Thank you for your valuable comment. We believe that gynecological surgeries performed using the hinotori™ SRS system can provide perioperative outcomes that are non-inferior to those achieved with the da Vinci® Xi system. To validate these findings, prospective studies with larger sample sizes are warranted. Based on the reviewers' feedback, we have added this information to the conclusion section. (Line 244-247). |

Round 2
Reviewer 1 Report
Comments and Suggestions for Authors
Thanks for revising the manuscript that I suggested.
Comments on the Quality of English Languagenothing particular